# Fetal to Neonatal Heart Rate Transition during Normal Vaginal Deliveries: A Prospective Observational Study

**DOI:** 10.3390/children10040684

**Published:** 2023-04-04

**Authors:** Yuda Munyaw, Jarle Urdal, Hege Ersdal, Matilda Ngarina, Robert Moshiro, Ladislaus Blacy, Jorgen E. Linde

**Affiliations:** 1Faculty of Health Sciences, University of Stavanger, 4036 Stavanger, Norway; 2Department of Obstetrics and Gynecology, Haydom Lutheran Hospital, Haydom P.O. Box 9000, Tanzania; 3Department of Electrical Engineering and Computer Science, University of Stavanger, 4036 Stavanger, Norway; 4Department of Anesthesia, Stavanger University Hospital, 4011 Stavanger, Norway; 5Department of Obstetrics and Gynecology, Muhimbili National Hospital, Dar es Salaam P.O. Box 65000, Tanzania; 6Department of Paediatrics, Muhimbili National Hospital, Dar es Salaam P.O. Box 65000, Tanzania; 7Research Department, Haydom Lutheran Hospital, Haydom P.O. Box 9000, Tanzania; 8Department of Obstetrics and Gynecology, Stavanger University Hospital, 4011 Stavanger, Norway

**Keywords:** fetal heart rate, neonatal heart rate, newborn heart rate, vaginal delivery, normal delivery, heart rate transition

## Abstract

Documentation of fetal to neonatal heart rate (HR) transition is limited. The aim of the current study was to describe HR changes from one hour before to one hour after normal vaginal deliveries. We conducted a prospective observational cohort study in Tanzania from 1 October 2020 to 30 August 2021, including normal vaginal deliveries with normal neonatal outcomes. HR was continuously recorded from one hour before to one hour after delivery, using the Moyo fetal HR meter, NeoBeat newborn HR meter, and the Liveborn Application for data storage. The median, 25th, and 75th HR percentiles were constructed. Overall, 305 deliveries were included. Median (interquartile range; IQR) gestational age was 39 (38–40) weeks and birthweight was 3200 (3000–3500) grams. HR decreased slightly during the last 60 min before delivery from 136 (123,145) to 132 (112,143) beats/minute. After delivery, HR increased within one minute to 168 (143,183) beats/min, before decreasing to around 136 (127,149) beats/min at 60 min after delivery. The drop in HR in the last hour of delivery reflects strong contractions and pushing. The rapid increase in initial neonatal HR reflects an effort to establish spontaneous breathing.

## 1. Introduction 

Most of the 140 million deliveries that occur globally every year are normal vaginal deliveries, and the majority take place with no identifiable complications to the mother, baby, or both at the onset of labor [1]. However, half of the stillbirths and three-quarters of neonatal deaths are reported to occur during labor and in the initial hours of life. Poor monitoring during labor is attributed to these deaths [2], and complications that arise in this period are not always predictable. For this reason, improving the quality of labor care, including heart rate (HR) monitoring, has been recommended as an important strategy to prevent deaths that occur around the time of delivery [2].

During the labor and delivery process, heart rate is commonly monitored to identify fetal and neonatal risks of hypoxia. Fetal heart rate (FHR) is frequently measured for the surveillance of fetal well-being before delivery [3]. Similarly, HR is immediately measured after delivery to assess neonatal well-being and is used to establish the need for and/or guide neonatal resuscitation [4].

Fetal heartbeats can be detected and monitored by ultrasound as early as five weeks of pregnancy [5]. Before 6 weeks, FHR is 100–115 beats per minute (bpm), after which it increases and peaks at 8 weeks to 144–159 bpm; by 9 weeks, it plateaus at 137–144 bpm [5]. Normal baseline FHR decreases slightly toward the end of pregnancy [6,7,8].

The pace of HR is controlled by the autonomic nervous system, baroreceptors, and chemoreceptors [9]. Some of the factors influencing FHR include sleep–wake patterns, breathing movements, medications, painful stimuli, sound and vibrations, and temperature. Moreover, maternal conditions during pregnancy, such as infections, may influence the fetal heart rate [10]. Drugs such as oxytocin, which are sometimes used for labor induction or augmentation, influence FHR by causing excessive uterine contractions [11]. As such, fetal HR during pregnancy and labor is influenced by different factors, making a description of a normal trace important in order to evaluate it and take clinical action if needed.

Guidelines derived from different studies and professional bodies are in agreement on recommending a baseline FHR of 110–160 beats per minute (bpm) during pregnancy and delivery [12]. However, during delivery, the first and second stages of labor are different in terms of physiology. During the second stage, physiological mechanisms such as frequent contractions and maternal expulsive efforts are dominant, and these may derange baseline FHR due to the compromise of fetal oxygenation [13]. In the second stage of labor during active pushing, the World Health Organization recommends fetal HR measurement every 5 min and through a contraction.

The set baseline FHR of 110–160 bpm needs further investigation, especially during the second stage of labor. Few old and recent studies have attempted to describe baseline HR changes that occur during the transition at normal delivery [14,15]. An old study described that HR undergoes wide fluctuations during the transition at delivery [14]. Additionally, a recent study evaluated FHR close to the time of delivery and showed that the baseline FHR decreases significantly toward the time of delivery [15]. However, the latest study did not take into account the mode of delivery, even though changes in FHR are known to influence the mode of delivery [16]. There is a need to specifically describe how FHR changes during the second stage and up to the moment of delivery in normal vaginal deliveries. This might enable healthcare providers to make confident decisions based on diverging from an expected normal FHR.

Studies describing HR changes just after birth have demonstrated a peak HR reached within one minute [17,18]. Factors such as mode of delivery, sleep–wake patterns, skin-to-skin care, gender, and body temperature have been shown to influence initial neonatal HR [19]. It has been further shown that the HR of neonates born vaginally is significantly higher up to one hour after birth than those born by Cesarean section [20]. No studies have documented the continuous HR course during the second stage up to delivery and immediately after delivery up to one hour. This is important to further understand the normal HR transition, as it may help care providers distinguish neonates who need resuscitation at delivery from those who do not need resuscitation.

It is estimated that up to 5% of term pregnancy babies may need assistance during the transition at birth. Additionally, healthcare providers in delivery rooms are sometimes faced with the dilemma of recognizing which babies need assistance and which ones do not need assistance. This may lead to unnecessary intervention or delay in necessary intervention. As HR is the most commonly used parameter to monitor fetal and neonatal well-being during the transition, it is important to further understand its course immediately before and after delivery. By increasing the knowledge on HR transition, healthcare providers will be in a better position to recognize babies who need assistance and those who do not. This will help to avoid unnecessary interventions and the associated complications, while at the same time, avoid delays in necessary intervention.

Therefore, the aim of this study is to describe the fetal-to-neonatal HR transition from one hour before to one hour after normal delivery in a cohort of uncomplicated vaginal deliveries with normal neonatal outcomes.

## 2. Materials and Methods

### 2.1. Settings

This study was part of the Safer Births [1] project, which aims to improve perinatal survival by gaining new knowledge and developing innovative products to improve care during childbirth. The study was conducted in the labor ward of Haydom Lutheran Hospital, a regional referral hospital in Tanzania. The labor ward provides comprehensive emergency obstetrics and neonatal care, and approximately 4000 deliveries are attended per year. Normal deliveries are mainly attended by midwives, backed up by registrars (medical doctors) and specialists (obstetricians). Fetal heart rate during labor is measured using a fetoscope or a doppler monitor called Moyo (Laerdal Global Health AS, Stavanger, Norway). Neonatal HR is measured and monitored using NeoBeat (Laerdal Global Health AS, Stavanger, Norway). Neonates under routine care stay with their mothers and, after one hour, both are transferred to the postnatal ward.

### 2.2. Participants’ Recruitment

The inclusion criteria were normal vaginal deliveries with neonatal outcomes of an Apgar score more than seven at five minutes and not ventilated. As the study aimed to describe HR transition in normal vaginal deliveries, we excluded stillbirths; gestational age below 37 weeks; multiple pregnancies; and those who had been induced, arrived late in the second stage of labor, undergone Cesarean sections, experienced cord prolapse, and suffered severe maternal bleeding. The recruitment of participants was carried out during admission to the labor ward, whereby the participants received information about the study and were asked for written consent by the admitting midwife. Upon consenting, the participants were enrolled into the study.

### 2.3. Data Collection

Quantitative data from the enrolled mothers and neonates were collected using a data collection form adapted from the Safer Births case report file. Trained research assistants filled in the required information through direct observation of deliveries and from the partogram. Variables included maternal age, gravidity, gestational age, birth weight, and Apgar score. HR data were collected using the Moyo FHR doppler monitor [2] and the NeoBeat neonatal HR meter [3] (Laerdal Global Health AS, Stavanger, Norway). The devices are shown in Figure 1.

Moyo was strapped to the mother’s abdomen throughout the second stage of labor until delivery. The second stage of labor is diagnosed once the cervix becomes fully dilated and ends with the delivery of the neonate, according to the World Health Organization. FHR was monitored and recorded continuously during the second stage. The serial number of the Moyo and the date and time of the monitor were recorded by the research assistants. After delivery, the research assistant extracted FHR doppler signal data from Moyo using the tablet-based Liveborn application (Laerdal Global Health AS, Stavanger, Norway). The Liveborn application is a research tool used for live observations of neonatal care during the first minutes after birth, and the research assistants were trained in its use.

Within a minute after delivery, neonates were placed on the mother’s abdomen and dried. Neobeat was placed on the chest before cutting the cord. A research assistant linking the two devices automatically transferred HR electrocardiography (ECG) signal data to the Liveborn application through a Bluetooth connection. The matched HR signal data from Moyo and NeoBeat in the Liveborn application were then uploaded to the Liveborn server.

### 2.4. Data Processing and Analysis

The dataset used in this work was from the Liveborn database and collected between 1 October 2020 and 30 August 2021. The FHR data from Moyo were cleaned using a previously proposed framework by Urdal et al. [15]. Segments of less than 30 s were removed, as the HR was considered unlikely to belong to the fetus, but rather the mother. More information on the method and chosen segment length to be removed can be found in the article by Urdal et al. [15].

The data analysis was performed using MATLAB 2021a, and the statistical analysis was performed in RStudio 2022.02.2+485 [21]. To illustrate the HR trend before and after birth, the median, 25th and 75th percentiles were calculated. To achieve a high resolution in the analyses, we calculated the median and percentiles based on HR observations after every 15 s. As a normal distribution of the FHR and neonatal HR cannot be assumed, the Wilcoxon rank test was used to determine if statistically significant changes in HR occurred. Characteristics describing the included mothers and babies are presented using median and interquartile ranges (IQR).

### 2.5. Ethical Clearance

The study received ethical clearance from the National Institute of Medical Research (NIMR) in Tanzania, reference number NIMR/HQ/R.8a/Vol. IX/3036 and the Regional Committee for Medical and Health Research Ethics, Western Norway (REK Vest reference number 2018–2408). The devices employed in the study are in routine use during deliveries at the hospital. Voluntary written informed consent was obtained from mothers during admission to labor. Care during delivery was provided according to hospital guidelines and mothers were treated equally regardless of consent status. For confidential purposes, all data were de-identified.

## 3. Results

A total of 3659 deliveries occurred during the study period, and 2205 were normal vaginal deliveries with normal neonatal outcomes. After the exclusions, 305 babies with HR of good signal quality were included, as shown in Figure 2.

The median maternal age was 24 (20–30) years, median gravidity was 2 (1–4), and gestational age was 39 (38–40) weeks. Neonates had a median birth weight of 3200 (3000–3500) grams and Apgar scores of 9 and 10 at the first and fifth min, respectively (Table 1).

The number of individual HR observations varied at different time intervals. The maximum was 198 individual observations recorded at 10 min before delivery and 303 recorded 3 min after delivery (Table 2).

The median time from the last FHR to delivery was 0.0 s (0.0, 15.0), and the median time from delivery to the first neonatal HR was 49.0 s (33.0, 71.0). A significant change in median HR was found from immediately before delivery (132) until 1 min after delivery (168) (*p* < 0.05). There was a significant drop in neonatal HR from 1 min (168) to 10 min (153) after delivery (*p* < 0.05), as shown in Figure 3.

A slight decrease in median FHR was observed in the last 60 min before delivery (from 136 to 131) (*p* = 0.4844) (Figure 4). The neonatal HR dropped from 168 at 1 min to 136 at 60 min after delivery (*p* < 0.05), as shown in Figure 4.

## 4. Discussion

In this study, we measured the HR of babies from one hour before to one hour after normal vaginal deliveries. The results indicate that FHR decreases in the last hour before delivery from a median of 136 to 131 bpm. This baseline heart rate falls within the established normal range (110–160), as documented in various guidelines [12]. In addition, the presented baseline FHR findings fall within category I in the National Institute of Child Health and Human Development classification system, which is strongly predictive of normal acid-base status when other parameters in the same category are normal [22]. This finding is expected considering that we only included uncomplicated normal deliveries in the study.

The slight decrease in FHR is explained by the physiological mechanisms of normal delivery, such as contractions, that result in a reduction in uteroplacental perfusion [9,23], causing transient fetal hypoxia. Additionally, maternal expulsive efforts and lying in a supine position frequently or constantly during pushing impair maternal breathing and blood flow toward the uterus [13]. These events, together or in isolation, impair fetal oxygenation, temporarily decreasing FHR, as shown by the results. The findings are different from those reported by Urdal [15], in which there was a significant decrease in FHR. While we only investigated uncomplicated normal deliveries, the study by Urdal investigated normal and complicated deliveries together, and this explains the difference in these findings.

In this study, all the neonates had normal outcomes, implying that they were able to withstand the labor stress in the last hour of labor. A healthy-term fetus with a normally developed placenta is able to overcome transient hypoxia by activation of the peripheral chemoreflex. This results in a prioritization of oxygenated blood to critical organs such as the heart, brain, and adrenals. Provided there is adequate time for placental and fetal reperfusion between contractions, the fetus is able to withstand intermittent hypoxia [13].

Clinical guidelines recommend no intervention in the case of a decrease in FHR during the second stage of labor, as long as the baseline is within 110–160 bpm and other parameters such as variability and late decelerations are in the normal physiological pattern. The interpretation should be in relation to the physiology of the second stage of labor, which differs from the first stage. When the decrease reaches less than 110, even if it is the second stage, it is important to be cautious of fetal hypoxia and to rule out causes of bradycardia such as uterine rupture, placental abruption, or umbilical cord prolapse. Once these life-threatening causes are ruled out, it is likely that the decrease is physiological and temporary. Then, a normalization of FHR should be expected after a short time. With these considerations, the risk of severe fetal hypoxia is less likely, and good outcomes should be expected in such labor. This knowledge can prevent unnecessary interventions during labor, and hence, reduce the risk of complications.

Our study further showed that neonatal HR significantly increases within one minute after delivery to 168 bpm, and further decreases significantly within one hour after delivery to 136 bpm. To our understanding, this is the first study offering new insight into how HR transitions from one hour before to one hour after normal delivery.

The first HR was recorded at a median of 49 s after delivery, and it reached a peak just before one minute. This concurs with findings from previous studies describing a rapid increase in normal neonatal HR after delivery, reaching a peak within the first minute [17,18]. The neonates included in our study started spontaneous air-breathing immediately after delivery, and this increased workload may partly explain the rapid initial increase in HR.

The results demonstrate that HR slowly decreased from its peak after one minute, with the trend continuing for 60 min, at which point HR was the same as 60 min before birth. Our results concur with a pulse oximetry-based study, which reported a similar decreasing HR trend up to one hour after birth [20]. The decrease can be explained by reduced stress after the neonate establishes spontaneous air breathing and skin-to-skin care is maintained. In research studies, neonates placed on the bare chest of their mother after delivery have been found to have lower cortisol levels at 60 min, and this likely reflects a reduced stress response and an associated reduced sympathetic drive of heart rate. In addition, skin-to-skin care calms the neonate and maintains their body temperature. Neonates kept skin-to-skin usually demonstrate a lower heart rate and respiratory rate, reflecting reduced stress.

Other factors that have been reported to influence initial neonatal HR include sleep–wake state, sex, and body temperature [19]. In addition, the timing of cord clamping has been reported to have an effect on initial neonatal HR. One study showed that neonates with delayed cord clamping had a lower HR than those with early cord clamping [24]. The neonates included in our study were in a sleep–wake state and skin-to-skin care during observations. Skin-to-skin care and delayed cord clamping are standards of care for normal vaginal deliveries in our study setting. These factors further underlie HR decrease after an initial rapid increase. It should be noted that HR is not the only important sign to be monitored during the transition. Other parameters, such as breathing/respiration, color, and grimacing, are equally important to be monitored for a smooth transition, as will be determined by Apgar scoring.

This is the first ECG-based study to show HR trends from the point of delivery up to 60 min after delivery. Previous ECG-based studies have demonstrated HR trends up to five minutes after delivery [17,18]. The use of dry-electrode ECG technology-based devices such as NeoBeat has made it possible to overcome challenges such as the prompt acquisition of HR signals immediately after delivery. This technology is known to enable the quick detection of neonatal HR; hence, it may support the decision of resuscitative measures and further care [25].

Our study was limited by uniform data availability along different time points, reducing the number of inclusions. Measurement bias might have occurred, for example, if Moyo recorded maternal HR instead of FHR. However, we excluded recordings that were suggestive of maternal HR. A mismatch of Moyo and NeoBeat data due to interrupted Bluetooth connections during signal data extraction contributed to the loss of data. The first measured HR occurred at 49 s after birth. This is later than has been shown by other studies which have utilized NeoBeat and been able to measure the first HR as early as 3 s after delivery. However, the peak HR detected within one minute was similar to the other studies using the same technology.

## 5. Conclusions

During the normal transition from intrauterine to extrauterine life, HR undergoes significant changes. The slight drop in FHR towards the time of delivery reflects strong uterine contractions and maternal pushing during the second stage. The rapid initial increase in neonatal HR reflects a stress response to the extrauterine environment after delivery and/or increased metabolic work due to the onset of spontaneous breathing.

## Figures and Tables

**Figure 1 children-10-00684-f001:**
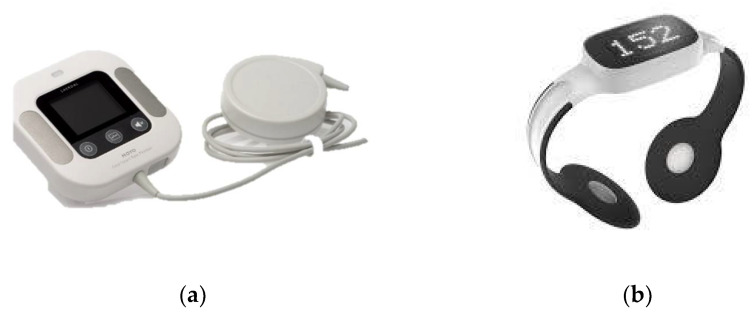
Moyo fetal heart rate monitor (**a**) and NeoBeat heart rate meter (**b**).

**Figure 2 children-10-00684-f002:**
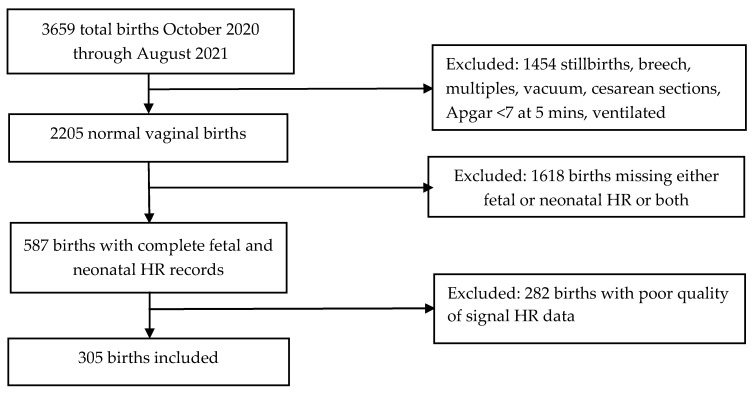
Flow diagram showing enrollment of participants. HR = heart rate.

**Figure 3 children-10-00684-f003:**
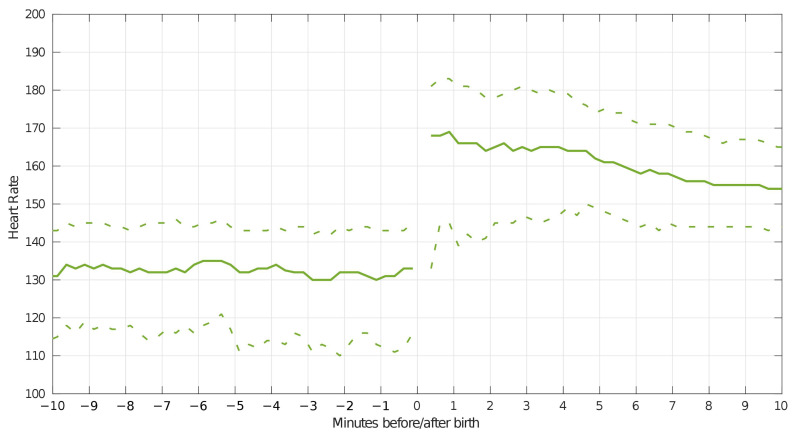
Illustration of heart rate transition 10 min before and after delivery. The solid line shows the median heart rate, and the dashed lines show the 25th and 75th percentiles. Each point on the curves indicates the median and percentile for a 15 s time segment.

**Figure 4 children-10-00684-f004:**
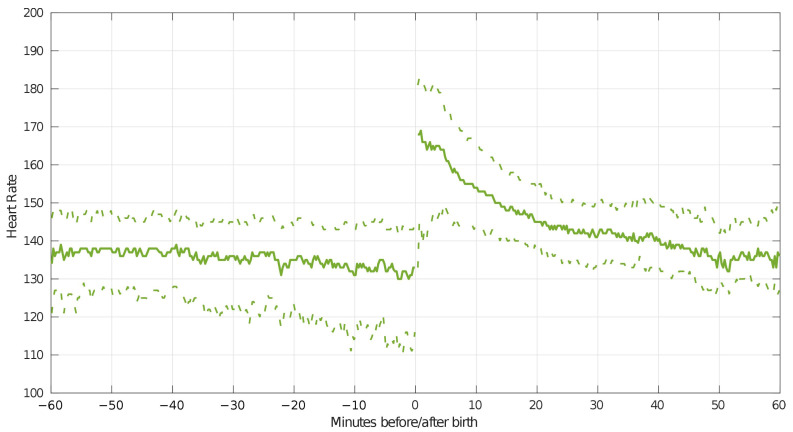
Illustration of heart rate transition from one hour before delivery to one hour after delivery. The solid line shows the median heart rate, and the dashed lines show the 25th and 75th percentiles. Each point on the curves indicates the median and percentile for a 15 s time segment.

**Table 1 children-10-00684-t001:** Characteristics of participants.

Characteristic	Median	Interquartile Range (IQR)
Maternal age in years	24	20–30
Gravidity	2	1–4
Gestational age in weeks	39	38–40
Birth weight in grams	3200	3000–3500
1st min Apgar score	9	9–9
5th min Apgar score	10	10–10

**Table 2 children-10-00684-t002:** Median heart rate at different time points and number of individual heart rate observations included.

Time (Minutes)	Fetal HR ^1^ in Beats/Minute—Individual Observations	Neonatal HR in Beats/Minute—Individual Observations
60	136 [125, 147]–127	136 [127, 149]–21
30	136 [123, 145]–194	143 [135, 150]–73
10	132 [114, 143]–198	153 [143, 163]–221
5	135 [119, 145]–194	160 [146, 174]–298
4	132 [113, 143]–193	163 [149, 176]–303
3	133 [114, 144]–190	165 [146, 179]–303
2	130 [112, 143]–190	165 [146, 180]–302
1	131 [114, 143]–187	165 [141, 180]–287
0	132 [112, 143]–183	168 [143, 183]–196

^1^ HR = Heart rate, presented as median (percentiles).

## Data Availability

Datasets may be available upon reasonable request to the corresponding author.

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
