# Peer review of "Fetal to Neonatal Heart Rate Transition during Normal Vaginal Deliveries: A Prospective Observational Study"

_children, 2023, doi:10.3390/children10040684_

Round 1

Reviewer 1 Report

Thank you for your submission about “Fetal to neonatal heart rate transitional assessment based ECG during normal vaginal deliveries.”. I think it will be better if you refer to the contents below and revise the manuscript.

Page 1

Abstract

Line 26

39 (20-30) : Please check the IQR.

Line 29-31

The drop in HR in the last hour of delivery may be due to strong contractions and pushing. The rapid increase of initial neonatal HR may reflect effort to establish spontaneous breathing.

→ In conclusion, please describe only the contents that can be known by this study rather than estimated contents.

 1. Introduction

 Line 49

resuscitation → neonatal resuscitation

 Page 2

In the introduction, you wrote a lot of content like a review paper. Please describe only the content that is directly related to this study.

 2. Materials and Methods

 Page 3

 Line 121

Figure 1: Moyo fetal heart rate monitor and NeoBeat heart rate meter

Figure 1: Moyo fetal heart rate monitor (a) and NeoBeat heart rate meter (b)

 Page 6

 Discussion

 In the discussion, please describe the contents related to the results of this study rather than the contents of a simple review.

 Page 7

 Line 245

delay cord clamp delayed cord clamp

 5. Conclusion

 In conclusion, please describe only the contents that can be known by this study rather than estimated contents.

Thank you.

Author Response

Reviewer 1:

Abstract Page 1 Line 26: 39 (20-30): Please check the IQR.

Thank you for noticing this error. The median gestational age and IQR are corrected and now read as 39 (38-40).

Page 1 Line 29-31

The drop in HR in the last hour of delivery may be due to strong contractions and pushing. The rapid increase of initial neonatal HR may reflect an effort to establish spontaneous breathing.

→ In conclusion, please describe only the contents that can be known by this study rather than estimated contents.

Thank you for your comment. We have revised the abstract conclusion as follows:  The drop in HR in the last hour of delivery reflects strong contractions and pushing. The rapid increase of initial neonatal HR reflects an effort to establish spontaneous breathing. Lines 29-31

1.Introduction Line 49 resuscitation → neonatal resuscitation

Page 2 In the introduction, you wrote a lot of content like a review paper. Please describe only the content that is directly related to this study.

Thank you for your addition. The word ‘‘neonatal’’ is now added to the sentence.

Thank you for your comment. During submission, there was a demand from the editorial office that we have to increase the content of our paper and word count, which is why we wrote a lot of content. We feel that most of the contents in the introduction are relevant to the study, however, there are few contents related to the background of the topic

2. Materials and Methods Page 3 Line 121

Figure 1: Moyo fetal heart rate monitor and NeoBeat heart rate meter

→ Figure 1: Moyo fetal heart rate monitor (a) and NeoBeat heart rate meter (b)

Thank you for the comment. The figures are now labeled as Moyo fetal heart rate monitor (a) and NeoBeat heart rate meter (b)

Page 6 Discussion

In the discussion, please describe the contents related to the results of this study rather than the contents of a simple review.

Thank you for this general comment. We feel that our discussion is in line with the study results, however we have added few sentences for more elaborations

Page 7 Line 245

delay cord clamp → delayed cord clamp

5. Conclusion

In conclusion, please describe only the contents that can be known by this study rather than estimated contents.                                                                

Thank you for the comment. The word delay is corrected and now read as delayed.

Thank you for your comment. We have revised the conclusion as follows: The slight drop in FHR towards time of delivery reflects strong uterine contraction and maternal pushing during the second stage. The rapid initial increase in neonatal HR reflects a stress response to the extrauterine environment after delivery and/or increased metabolic work due to the onset of spontaneous breathing. Lines 269-272

Reviewer 2 Report

Dear author,

This is a well written manuscript regarding the transition in HR from fetal life to the extrauterine life. I have some comments to make.

In line 26 of the Abstract you report Median gestational age 29 weeks and IQR 20-30. Is the IQR correct? The 75th percentile is smaller than the Median value.

The followed methodology, and statistical analysis are both sound. Nevertheless, from the Methods section the definitions of delivery stages must be added. 

In line 156 I do not understand the phrase "voluntary written consent". If they did not give informed concent their babies weren't included in the study?

In line 164 the Median (IQR) for gestational age is not in accordance with that reported in the Abstract. 

Figure 3, which depictes the transition in FHR from 10 minutes before delivery to NHR up to 10 minutes after delivery is very helpfull for someone to grasp the reported results. The same with figure 4.

The Discussion is well written according to the study findings.

Regarding the Strengths and Limitations of this study, it is the first that meassures HR 60 minutes before to 60 minutes after delivery, but does this study add something more than the previous ones that meassured HR for shorter periods of time?

The Conclusions are in accordance to the results.

Author Response

Reviewer 2:

This is a well-written manuscript regarding the transition in HR from fetal life to extrauterine life. I have some comments to make

Thank you for reading our paper and comments.

In line 26 of the Abstract, you report Median gestational age of 29 weeks and IQR of 20-30. Is the IQR correct? The 75th percentile is smaller than the Median value.

Thank you for your good observation. The sentence is revised and now read as follows in lines 25-26: Median (interquartile range; IQR) gestational age was 39 (38-40) weeks

The followed methodology and statistical analysis are both sound. Nevertheless, from the Methods section, the definitions of delivery stages must be added

Thank you for your comment. We have revised and added the following sentence to the methods section lines 122-123 to define the second stage as used in this study. ‘‘The second stage of labor was diagnosed once the cervix becomes fully dilated and ended with the delivery of the neonate’’

In line 156 I do not understand the phrase "voluntary written consent". If they did not give informed consent their babies weren't included in the study?

Thank you for the comment. We obtained ‘‘informed written consent’’ from the mothers to record the heart rates of fetuses and neonates as described in lines 157.

In line 164 the Median (IQR) for gestational age is not in accordance with that reported in the Abstract. 

Thank you for this observation. What is reported in line 164 is maternal age and what is reported in the abstract is gestational age. We didn’t include maternal age in the abstract due to the words limit

Figure 3, which depicts the transition in FHR from 10 minutes before delivery to NHR up to 10 minutes after delivery is very helpful for someone to grasp the reported results. The same with figure 4.

Thank you for looking at the figures and the compliment.

The Discussion is well written according to the study findings.

Thank you for your compliment.

Regarding the Strengths and Limitations of this study, it is the first that measures HR 60 minutes before to 60 minutes after delivery, but does this study add something more than the previous ones that measured HR for shorter periods of time?

Thank you for this comment. This study helps to show more on HR during the transition for extend period compared to others. To clinicians this finding might serve as reference HR up to one hour before and after birth.

The Conclusions are in accordance to the results

Thank you for reading and commenting.

Reviewer 3 Report

 It is a well-written article with a novel hypothesis.

1.    I believe that the results could be even more robust if “heart rate in neonatal period” was not the only “sign”.  It would be interesting if the authors give more information on the discussion related to its use in everyday practice.  

 2.    There is not enough comment on the use of specific devices. what is this based on? Is there a reference?

3.    Line 41: In the introduction:  “complications that arise in the course of labor are not predictable” would substitute with “complications that arise in the course of labor are not always predictable”

 4.    Line 95: Please rephrase as obstetricians are medical doctors 

5.    The discussion has to be more specific and informative.

6.     It would be better for the authors to clarify the “Inclusion criteria were normal vaginal deliveries with neonatal outcomes of Apgar score of more than seven at five minutes and not ventilated”, in order to be more informative for the readers.

 Author Response

Reviewer 3

I believe that the results could be even more robust if “heart rate in neonatal period” was not the only “sign”.  It would be interesting if the authors give more information on the discussion related to its use in everyday practice.  

Thank you for this important comment. It is true that HR is not the only sign for neonatal wellbeing. Other signs such as crying, breathing/respiration, color and muscle tone are equally important. However, HR is commonly used to guide resuscitations of non-breathing newborns and monitoring thereafter. This makes it unique and central in newborn monitoring. We have added a sentence in Lines 243-245 for elaboration

There is not enough comment on the use of specific devices. what is this based on? Is there a reference

Thank you for your comment. The devices used to measure heart rate are used in daily clinical practice routinely as described in lines 156-157. This study was not focused on using specific devices but rather uses what was available to collect heart rate data. However, we have explained about the devices in lines 116-117 and the link for reference.

 Line 41: In the introduction: “complications that arise in the course of labor are not predictable” would substitute with “complications that arise in the course of labor are not always predictable”

Thank you for your comment. We have taken the comment and the word ‘’always’’ is added to the sentence. Lines 41-42

 Line 95: Please rephrase as obstetricians are medical doctors

Thank you for your comment. In the study settings medical doctors and obstetricians have different training and scope of practice. Medical doctors (MD) sometimes called registrars are general practitioners who completed their undergraduate medical degree while Obstetricians have undergone specialist master’s training in obstetrics and gynecology after completion of undergraduate medical degree. Revised Lines 95-96 for clarity

The discussion has to be more specific and informative

We have revised the discussion by adding some sentences to make it more informative

It would be better for the authors to clarify the “Inclusion criteria were normal vaginal deliveries with neonatal outcomes of Apgar score of more than seven at five minutes and not ventilated”, in order to be more informative for the readers.

 Thank you for your comment. The study inclusions were further clarified by a description of exclusion criteria as stated in lines 104-107: we excluded stillbirths, gestational age below 37 weeks, multiple pregnancies, inductions, arrived late in the second stage, cesarean sections, cord prolapse, and severe maternal bleeding. The study flow chart (figure 2) on page 4 further illustrates the inclusion and exclusion process further.

Round 2

Reviewer 2 Report

Dear authors,

I would like to thank you for responding to all my comments.

I would also like to congradulate once again for your excellent study.

I am looking forward to reading the published version.